# Later but Not Weaker: Neural Categorization of Native Vowels of Children at Familial Risk of Dyslexia

**DOI:** 10.3390/brainsci12030412

**Published:** 2022-03-21

**Authors:** Ao Chen

**Affiliations:** 1School of Communication Sciences, Beijing Language and Culture University, Beijing 100083 China; aochen@blcu.edu.cn; 2Utrecht Institute of Linguistics, Utrecht University, 3512 JK Utrecht, The Netherlands

**Keywords:** phonological categorization, infants, familial risk of dyslexia, mismatch negativity

## Abstract

Although allophonic speech processing has been hypothesized to be a contributing factor in developmental dyslexia, experimental evidence is limited and inconsistent. The current study compared the categorization of native similar sounding vowels of typically developing (TD) children and children at familial risk (FR) of dyslexia. EEG response was collected in a non-attentive passive oddball paradigm from 35 TD and 35 FR Dutch 20-month-old infants who were matched on vocabulary. The children were presented with two nonwords “*giep*” [ɣip] and “*gip*” [ɣIp] that contrasted solely with respect to the vowel. In the multiple-speaker condition, both nonwords were produced by twelve different speakers while in the single-speaker condition, single tokens of each word were used as stimuli. For both conditions and for both groups, infant positive mismatch response (p-MMR) was elicited, and the p-MMR amplitude was comparable between the two groups, although the FR children had a later p-MMR peak than the TD children in the multiple-speaker condition. These findings indicate that FR children are able to categorize speech sounds, but that they may do so in a more effortful way than TDs.

## 1. Introduction

Developmental dyslexia (DD) is an impairment of reading and spelling skills despite normal intellectual abilities and educational opportunities [1]. The estimates of prevalence of dyslexia vary from 3 to 10%, depending on measures and inclusion criteria. There is wide agreement that dyslexia has a genetic basis [2,3,4], and children with a dyslexic parent have a 29–66% risk of developing dyslexia [5]. Even though a large proportion of children at family risk (FR) do not develop dyslexia, they still perform more poorly than typically developing (TD) children on tasks such as spelling, non-word reading, and reading comprehension [6,7,8].

There is wide consensus that phonological awareness, namely, the ability to manipulate, generate or judge sound units, such as syllables, onsets, rhymes or phonemes, serves as the basis for script decoding and reading [9,10,11]. Phonological awareness difficulties, particularly phoneme awareness difficulties, are deeply rooted among people with dyslexia, and they do not disappear even after reading difficulty has been largely compensated [12,13,14].

Atypical speech perception has been hypothesized to underlie phonological awareness impairment among children with dyslexia [15,16,17,18,19]. When studying DD, speech perception has often been operationalized as categorical perception of phonemes. In categorical perception experiments, listeners are presented with stimuli that gradually change from one phonological category to another in a stepwise fashion, for instance, from *pea* to *bee*. Typical listeners perceive such gradual changes as having distinct categories (i.e., identification shifts from *pea* to *bee* consistently at one specific step) and they discriminate the between-category stimuli better than the within-category stimuli. The essence of categorical perception is that listeners disregard lexically irrelevant acoustical differences while attending to differences of the same magnitude if they change word meaning. Categorical perception is essential for efficient speech processing since variability abounds in human speech. The same phoneme or word can drastically differ acoustically due to speakers’ anatomical structures, speaking rate, and phonetic context, etc. [20,21,22]. Nevertheless, adults are able to recognize linguistic content in the face of variation, and we disregard linguistically irrelevant acoustical differences while attending to the phonemic differences that contrast (lexical) meaning. Grapheme–phoneme mapping, the backbone of reading, is only possible if phonemes can be represented categorically [23,24].

It continues to be debated whether the whole group or only a subgroup of children with DD (or with a high risk of DD) show atypical categorical perception of phonemes. On the one hand, multiple studies have shown that children with DD have fussier categorical boundaries of phonemes [25,26,27,28,29,30,31], and preschool speech perception has been found to be a predictor of reading, independent of phonological awareness, and speech perception of children with DD was found to be significantly delayed compared to TD children [32]. In the temporal sampling framework [33], Goswami proposed that children with DD may exhibit less efficient Theta phase locking, leading to impaired syllable-level processing, yet their phonetic-level Gamma sampling network may remain unimpaired, resulting in over-sensitivity to phonetic contrasts. On the other hand, other studies failed to find convincing evidence supporting impaired categorical perception among children and adults with DD, and quite often only a subgroup of children with DD show atypical speech sound perception [26,34,35,36]. Therefore, it remains unclear whether atypical categorical perception can be considered as a labeling feature for DD that applies to the whole group or whether it only applies to a subgroup of people with DD.

Importantly, deviations in speech perception in dyslexic and FR children may occur prior to formal instruction in reading and writing [37] and these might be precursory to dyslexia. For example, using a conditioned head-turn procedure, the authors of [38] showed that by six months the FR children took significantly longer to categorize native intervocalic consonants than the TD children, indicating attenuated speech sound categorization among the FRs. In the first year of life, when tested with a habituation–dehabituation procedure, FR infants failed to exhibit improvement in the discrimination of native vowel contrast as the TD infants did [39]. In preschool years, FR children have been found to perceive native phonemes less categorically than TDs [40,41,42,43]. It should be noted, however, that in the above-mentioned behavioral studies, the intra-token variability was barely controlled. For example, the contrasts (i.e., /faːp/, /feːp/, /sæn/, and /sɛn/) were produced by four female speakers in [39], while in [37] one single token of *ata* was used to generate the continuum from *ata* to *atta*. Without a control condition where only single tokens of each word were used as stimuli, it is difficult to ascertain whether FRs were less able to discriminate phonetic/acoustical differences without the presence of inter-token variation or they were less capable of variability normalization compared to TDs.

Neural signatures of speech sound discrimination have been widely studied with mismatch negativity (MMN) [44,45], a component of auditory event-related potentials (ERPs). MMN can be elicited using a passive oddball paradigm, in which listeners are presented with a stream of repeating ‘standard’ sounds conforming to a certain regularity punctuated occasionally by ‘deviant’ sounds, dissimilar in some relevant dimension from the standards. If the brain detects the change from standard to deviant, then on the difference waveform obtained by subtracting the response to the standard from that to the deviant, the MMN is visible as a negative peak between 100 and 300 ms from deviant onset in adults [46,47]. Besides physical difference between the standard and deviant, MMN can also be elicited by violation of abstract patterns, and listeners are able to extract similarity from the standard and deviant and detect dissimilarity across the two types [48,49,50,51]. MMN has been found to be sensitive to the magnitude of stimulus change, and its amplitude gets larger and peak latency shorter as the magnitude of deviation increases [47]. In addition, MMN has been found to correlate with behavioral discrimination accuracy [52].

MMN has been commonly used to understand auditory change detection among children, yet its interpretation is hampered by the fact that under certain conditions and at certain ages, infants’ and children’s mismatch responses tend to exhibit a late *positivity* (positive mismatch response, p-MMR) rather than adult *negativity* (MMN) [53,54,55,56]. As infants develop, the polarity of the mismatch (MMR) shifts to negative, and gradually approximates the adult MMN [56,57]. Yet no consistent results have been found with regard to the age at which infant/child p-MMR shifts to MMN [54,55,56,58]. It has been hypothesized that the shift from p-MMR to MMN would occur earlier for acoustically salient stimuli (i.e., stimuli showing greater physical difference) than for non-salient stimuli [55,59,60].

Atypical MMN has been commonly reported for people with DD [19,61], and a recent meta-analysis has shown that, irrespective of age, speech-elicited MMN is aberrant among people with DD [62]. DD and FR children have been reported to exhibit later MMN peak latencies, possibly reflecting more effortful processing of speech sounds [63,64,65]. Although differences in speech-elicited MMN between TDs and people with DD can be observed before literacy education (e.g., [66]) and neonatal brain responses have been found to be predictive of DD [67], findings on how the MMN of FR children may differ from that of TD children are inconsistent [68]. Atypical MMN prior to literacy education, such as reduced MMR amplitude and different MMR scalp distribution, has been reported among German, Italian, and Chinese FR children [69,70,71]. MMN in FR children was extensively studied in two longitudinal projects, the Jyvaskyla Longitudinal Study of Dyslexia (JLD) conducted with Finnish learning infants in Finland [72] and the Dutch Dyslexia Project (DDP) conducted with Dutch learning infants in the Netherlands [37]. In the JLD, TD children and FR children were followed from birth through adulthood, and MMN was elicited by consonant and vowel duration changes as well as non-speech sounds. JLD has shown that atypical ERPs to speech sounds after a few days of birth were the first precursors of dyslexia [72]. By six months, when the standard and the deviant differed in terms of consonant duration, the FR group failed to exhibit the adult-like MMN that was observed among the TD infants [73]. When presented with vowel duration difference, p-MMR had a larger amplitude among the FR and the TD newborns [74]. In another recent study, the MMR elicited by vowel change (i.e., /ta/–/to/) showed comparable amplitude between FR and TD Finnish learning newborns, yet the MMR was more right-lateralized among the TDs [75]. Taken together, whether and how MMN differs between FRs and the TDs before literacy education remains undetermined. The between-group difference, when present, may exhibit as latency, amplitude, or lateralization difference.

For the research with Dutch infants, it has been found that those FR infants who later developed DD failed to show mismatch response when presented with a native consonant change /bAk/–/dAk/ [76,77,78]. Hakvoort et al. (2015) [79] showed that although FR dyslexics, FR non-dyslexics, and control children all showed MMN to changes in amplitude rising time and frequency, FR children showed attenuated MMN to intensity change, and they exhibited different MMN lateralization to frequency change compared to the TD children. Van Leeuwen (2008) [77] made use of two tokens in the middle of a continuum changing from /bAk/ to /dAk/, each belonging to a different category (i.e., either /bAk/ or /dAk/) and found that two-month-old FRs showed attenuated and less left-lateralized MMR compared to the TD infants. In another study that tested neural discrimination of /bə/ and /də/, Noordenbos and colleagues [65] found that when the standard and the deviant stimuli straddled the phonemic boundary, MMN was elicited in 6-year-old TD and FR children, yet it was less prominent among the FRs, and the FR children showed a later MMR peak than the TD children. However, when the standard and deviant belonged to the same phonemic category, only the FR but not the TD children showed MMN. Taken together, these studies with Dutch FR children indicated that MMN to some but not all phonetic difference can be attenuated among the FR children, and their brain response may be over-sensitive to allophonic variations [30].

In sum, although multiple studies found that auditory ERPs differed between TD and FR children and that such differences could be predictive of later reading [72], whether and how *discriminative* auditory ERPs or MMR may differ between TD and FR children is still unresolved. Inconsistently, mismatch response has been found to differ in terms of amplitude or scalp distribution between the groups. The group effect seems to be susceptible to the way that the stimuli were presented and characteristics of the speech sound (e.g., duration, intensity, frequency, or vowel category difference). The fact that both the Finnish and the Dutch studies made use of single standard and deviant made it difficult to ascertain whether the FR children responded to the phonetic/acoustical or categorical difference between the two types of stimuli.

Perhaps the only study that made use of variable standards and deviants is [80]. They tested adults with DD on the /æ/–/i/ contrast, and in the variable condition the standards were variable in terms of fundamental frequency (f0), while f0 remained constant in the constant condition. They found that the adults with DD showed comparable MMN to the TDs in the f0 constant condition but attenuated MMN in the f0 variable condition. These findings suggest that adults with DD were less capable than the TDs in native vowel categorization, which is consistent with the allophonic hypothesis of speech processing of DD.

To better understand whether FR and TD children categorize native phonemes differently at an early age, the current exploratory study investigated neural signatures of speech sound discrimination among 20-month-old TD and FR infants with MMN. We used native acoustically non-salient vowels, *giep* [ɣIp] and *gip* [ɣip], as the stimuli, since the non-salient contrasts have been found to be particularly challenging for children with DD [81]. In particular, to understand whether the two groups differed in phonetic discrimination or phonological categorization, two experimental conditions were designed. In the single-speaker condition, one token of [ɣIp] and one token of [ɣip], both produced by the same speaker, were used as stimuli, while in the multiple-speaker condition, twelve tokens of each nonword, each produced by a different speaker, were used as stimuli. Since the stimuli were obtained by manipulating natural production, in the multiple-speaker condition the tokens differed in terms of melodic contour (f0), voice, and phonetic implementation of the vowels. In essence, speaker normalization and classical categorization experiments require the same skill, namely, that listeners attend to acoustical differences that signal phonologically relevant speech contrasts while disregarding signals irrelevant to linguistic content. Indeed, findings on speaker normalization are similar to those obtained in classical categorical perception experiments, namely, that children with dyslexia were less able to normalize speaker variability [82]. Therefore, if the FR children were impaired in phonetic discrimination in general, their MMN should be less pronounced than the TDs in both the conditions, while if they were over-sensitive to allophonic variations [83], the between-group difference should be more evident in the multiple-speaker condition.

We focused on 20 months of age, since children’s vocabulary quickly expands at this stage and the need to differentiate similar sounding words is a driving force for refining phonological representation [84,85]. It has been reported that 20-month-old FR children had significantly smaller vocabularies compared to TD children [86]. Thus, it is plausible that at this age, deviant word learning and atypical phonological representation correlate with each other. To identify the influence of at-risk status and remove the possible vocabulary effect, we paired each individual FR child with a TD child closely matched for vocabulary, using the Dutch version of the MacArthur–Bates Communicative Development Inventory: Words and Sentences (N-CDI) [87]. A total of 702 words are listed in the N-CDI and they are divided into 22 different semantic categories (i.e., animals, vehicles, toys, food and drink, clothing, body parts, small household items, furniture and rooms, action words, descriptive words, pronouns, question words, prepositions and locations, quantifiers and articles, helping verbs, connecting words, sound effects, items outside the house, places outside the house, people, games and routines, and words about time). For each listed word, the parents were asked to indicate whether their child “understands” or “understands and produces” it.

Vowels were used as stimuli since these have been widely studied in speaker normalization and categorization research. Vowels are mainly defined by their formants or resonant frequencies as the result of a particular vocal tract configuration. First (F1) and second (F2) formants are the primary acoustic determinants of vowels, with the former reflecting vowel height and the latter vowel backness. The adult brain is able to separate phonologically contrastive vowel categories in the face of speaker and within-category variability [88,89,90,91].

## 2. Methods and Materials

### 2.1. Participants

Thirty-nine FR children participated in the research. Three were excluded due to excessive head movement, and one was excluded for missing N-CDI. Fifty-nine 20-month-old TD infants participated in the current study, and six were excluded from analysis due to crying or excessive head movements. From the 53 remaining infants, for each FR child, a TD child of the same gender, with the most similar N-CDI productive score (i.e., the total number of words listed as “understands and produces” by the parent), and the most similar age was matched. A total of 70 children (35 TDs and 35 FRs) were included in the final analysis.

Table 1 lists the information about the TD and FR children included in the current study. No impaired hearing or other cognitive delays in the children were reported by any of the parents. When the productive scores of all the participants were compared (i.e., the 53 TDs and the 35 FRs), univariate ANOVA, with group being the independent variable, found no significant difference between the groups, F(1, 86) = 2.52, *p* = 0.12, with the TDs having a mean (SD) productive score of 133 (112) and the FRs a score of 98 (84).

The children were labeled as FR if at least one of the parents was reading-impaired (phonological dyslexia), which was determined by three tests administered at either the Utrecht or Groningen labs. Two of them were reading tests, namely, the ‘Een-Minuut-Test’ (EMT; [92]) and the ‘Klepel’ [93]. In the EMT, a parent was asked to read out loud a list of real words as quickly and accurately as possible within one minute. The real words differed in frequency, and for the words with very low frequency, phonological decoding was needed for successful reading. In the Klepel test, the parents were instructed to read out loud a list of pseudowords as fast as possible within two minutes, and the nonwords were spelled in such a way that they obeyed the grapheme–phoneme mapping regularity. The other test was the Analogies subtest of the comprehension subscale of the Dutch version of the Wechsler Adult Intelligence Scale [94]. A parent was classified as reading-impaired if he/she had a score at the lowest 10% in one of two reading tasks, or at the lowest 25% in both, or he/she had a discrepancy greater than 60% between a high score on the WAIS Analogies subscale and the score on one of the reading tests. The last criterion was included to identify highly educated parents who were competent in verbal intelligence and who had received large amounts of reading and spelling training but whose reading ability was still low compared to their verbal intelligence [95].

### 2.2. Materials

Twelve female native Dutch speakers (mean age = 21 years, SD = 2.5 years) were recruited to produce the stimuli. They were first visually familiarized with a list of printed CVC nonword minimal pairs, where the two nonwords in a pair were solely distinguished by the vowel (e.g., tos [tos] and toes [tus], nief [nif] and nif [nIf]). They could spend as much time as they needed to read and get familiar with these nonwords. The target nonwords for the current experiment, *giep* [ɣIp] and *gip* [ɣip], were one of these pairs. The rest of the nonwords were for a word-learning experiment which together with the current experiment formed parts of a larger project examining early phonology and word development. Next, the participants were asked to produce all the nonwords in carrier sentences *ik zei niet X maar ik zei Y* as well as *ik zei niet Y maar ik zei X* (meaning I did not say *X* but I said *Y,* and vice versa), where *X* and *Y* were one of the minimal pairs. They were told to speak the sentences as if they were talking to a toddler. The speakers were recorded with a Sennheiser ME-64 microphone and a DAT recorder TASCAM DA-40 in a sound-attenuated room.

For each speaker, one well-realized token of [ɣIp] and one of [ɣip] were cut off from the recording for further manipulation in PRAAT [96]. All the tokens had a falling f0 contour. The duration of the tokens was manipulated to have a mean of 344 ms (SD = 9.5 ms, range 323–361 ms) and the intensity was scaled to 70 dB. These manipulated [ɣip]s and [ɣIp]s were used as stimuli in the current experiment. All the twelve tokens of each nonword were presented in the multiple-speaker condition, while the same tokens of each nonword from one speaker were presented in the single-speaker condition for all the participants. Duration and mean f0 were measured for the vowel part (i.e., /i/ and /I/), and F1, F2, and F3 values were measured at the temporal midpoint of the steady part of the vowels. A paired *t*-test was conducted with each of these measurements to examine the difference between the two vowels. Table 2 lists the mean values of these measurements and the results of the *t*-tests. Figure 1 plots the F1 and F2 values of each individual vowel in the stimuli. The [ɣIp] and [ɣip] used in the single-speaker condition had an F1 of 348 Hz and 485 Hz, an F2 of 2827 Hz and 2361 Hz, and an F3 of 3432 Hz and 3100 Hz, respectively. As can be seen, the acoustical characteristics of the vowels in the stimuli were consistent with those reported in previous studies [97]. Multiple native Dutch adult speakers listened to the stimuli and reported the stimuli to be natural, and all were able to identify all the stimuli as [ɣIp] or [ɣip] correctly.

We used CVC nonwords rather than isolated vowels as stimuli because /i/ can occur as a reduced format of *hij* (meaning he) in colloquial Dutch, as in the sentence *wat doet ie* (meaning *what does he do*), whereas /I/ alone can never be a word. Using nonwords as stimuli precluded lexical status from being a confounding factor. Although the Dutch /i/ and /I/ may be considered to contrast in duration besides F1/F2, acoustical analysis has shown that duration does not distinguish these vowels sufficiently [97,98]. Dutch, however, does have long and short vowels, as in *maan* [ma:n] (meaning moon) and *man* [mɑ] (meaning man), and both Dutch adults and infants were found to be sensitive to long vowels being mispronounced as short ones but not vice versa [99,100]. Therefore, to prevent duration from being a confounding factor, the vowels in [ɣIp] and [ɣip] were not manipulated to contrast in duration.

In both conditions, [ɣip] was the standard and [ɣIp] was the deviant. It should be acknowledged that, according to previous studies on vowel perception, to detect a change from a less to a more peripheral vowel is easier than the other way around [101,102], hence the assignment of standard and deviant may have an influence on mismatch response. Yet it was not our purpose to investigate asymmetry in neural detection of the vowel change. In addition, it is practically more feasible to match the TD and FR children with a consistent assignment of standard and deviant across the participants.

### 2.3. Procedure

A passive oddball paradigm was adopted. Infants’ brain responses were recorded in two blocks: a multiple-speaker block followed by a single-speaker block. Since the current study focused on whether the FRs differed from the TDs on speaker normalization, the multiple-speaker block always preceded the single-speaker condition. Each block comprised 600 stimuli, of which 480 (80%) were standards and 120 (20%) deviants. Each block began with 10 repetitions of the standard, after which standards and deviants were presented in a pseudo-random order with the constraint that deviants were separated by at least two standards. The inter-stimulus interval (ISI) was randomly varied between 320 ms and 400 ms.

The EEG was recorded in a sound-attenuated room in the Institute of Linguistics at Utrecht University. The infant participants sat on their caregivers’ laps during the experiment. Infant-friendly silent animated videos were played on the computer screen, and parents were instructed not to talk during the experiment. Toys were placed on the table in front of the infant, with which they could play if they wanted to. The distance between the participant’s eyes and the screen was ~1 m and the experimental stimuli were presented at 70 dB SPL (measured from where the infant sat) through two audio speakers on each side of the screen. EEG was recorded with a Biosemi system from a 32-channel cap with Ag/AgCL electrodes according to the 10–20 International System of Electrode Placement. EEG was recorded at a sampling rate of 1024 Hz.

The infants’ word knowledge was measured with N-CDI. The parents filled in the N-CDI at home online, either before or after the experiment. For each word, they were asked to indicate by mouse-clicking whether their child “understands but does not produce yet” or “understands and produces”. The raw and percentile scores were automatically generated with locally developed software.

### 2.4. EEG Processing

The EEG data were analysed offline using EEGLAB toolbox (version 13.1.1b in Matlab 2015b, [103]). The raw recordings were down-sampled to 250 Hz and filtered between 0.3–20 Hz. The continuous recordings were re-referenced to the average of all electrodes and segmented into 700 ms epochs from 100 ms before the onset (baseline) to 600 ms after the stimulus onset. Continuous bad channels were visually inspected and interpolated. Twenty-seven participants had channels interpolated, and on average 0.96 (SD = 0.88) channels was interpolated. Trials having an amplitude larger than ±150 microvolts were removed. The standards immediately after a deviant were excluded from analysis. The remaining artefact-free trials were averaged to obtain the ERPs for each infant. Infants who had more than 50 artefact-free deviant trials were included in the final analysis, and a further two infants were excluded. Individual waveforms of the remaining 34 TD and 34 FR children were averaged to obtain the grand averaged waveform.

### 2.5. Statistical Analysis

As there has been ample evidence that MMR is most evident at frontal central scalp locations, analysis was conducted with latency and amplitude measurements obtained at F3, Fz, F4, C3, Cz, and C4 [65]. For the TD and FR group separately, to identify the onset and offset of the MMR (if any), for each condition, point-by-point *t*-tests were performed with the standard and deviant ERPs (i.e., ERPs of individual participants) for all the points between 200 and 600 ms after the stimulus onset. If for at least one electrode, the standard and deviant ERPs significantly differed at a minimum of six consecutive time points (i.e., 24 ms, with the sampling rate being 250 Hz), the difference was considered meaningful [59], [104] and an MMR peak was subsequently identified on the grand average. Then individual MMR peak latencies were identified in the 100 ms window (50 ms before and after) surrounding the grand average peak, and individual MMR peak amplitudes were calculated as the mean amplitude in the 40 ms (20 ms before and after) window surrounding the individual peaks. To investigate how MMR differed across the groups, for each condition, repeated measure ANOVAs, with electrode (F3, Fz, F4, C3, Cz, C4) being the within-subject variable and group (FR and TD) being the between-subject variable, were conducted with individual peak amplitudes.

The significance of the MMR responses was also tested using non-parametric cluster-based mass permutation tests [105], implemented in the Fieldtrip toolbox in Matlab [106]. This analysis was completely data-driven and included all electrodes and all time points between 0 and 600 ms. First, a series of *t*-tests was computed for each electrode and at each time point. Then, clusters were formed over space by grouping electrodes (at least two adjacent electrodes) that had significant initial *t*-test results (*p* < 0.05) at the same time point. Clusters were formed over time by grouping adjacent time points that had significant *t*-values (*p* < 0.05). The sum of all *t*-values within each cluster provides a cluster-level *t*-score (mass *t*-score). A permutation approach was used to control for type I errors. For this the standard and deviant waveforms were randomly swapped and the *t*-tests were repeated 1000 times to generate a data-driven null hypothesis distribution. The observed *t*-values from the first step were compared with the null-hypothesis distribution. The cluster was considered significant if the mass *t*-score fell in the top 2.5 or bottom 2.5 percentile of the distribution. The cluster permutation statistics approach yields a conservative measure and there is a trade-off between sensitivity to local strong effects versus sustained smaller effects which are diffused across scalp locations [107].

### 2.6. Results

Table 3 lists TD and FR mean (SD) accepted trials in the multiple-speaker and single-speaker condition. A MANOVA, with the number of accepted trials being the dependent variable and group being the independent variable, found no significant difference between the two groups. F_multiple standard_ (1, 66) = 0.37, *p* = 0.55, F_multiple deviant_ (1, 66) = 0.52, *p* = 0.47, F_single standard_ (1, 66) = 1.15, *p* = 0.29, F_single deviant_ (1, 66) = 1.65, *p* = 0.20. The time windows where the standard and deviant ERPs differed significantly for each condition and each group are listed in the Appendix A, as are the mean peak amplitudes of the standard ERP, deviant ERP, and difference wave. Figure 2 plots the standard ERPs, the deviant ERPs, and the difference waves in the multiple- and single-speaker conditions of the TD and FR toddlers. Figure 3 plots the individual peak amplitudes.

As can be seen from Figure 2, in all the conditions, a p-MMR was observed. For the TD children, the p-MMR had a grand average peak latency of 344 ms and 356 ms in the multiple- and single-speaker condition respectively. The FR children had a grand average peak latency of 396 ms in the multiple-speaker condition and 354 ms in the single-speaker condition.

For the p-MMR peak latency measurements, for the multiple-speaker condition, univariate ANOVA, with group as the independent variable, showed that the effect of group was significant, F(1, 66) = 36.50, *p* < 0.001, partial η^2^ = 0.36, with the FR group having a higher mean p-MMR (i.e., mean of individual peak latencies) peak latency of 391.65 ms (SD = 30.61 ms) than the TD group’s latency of 346.71 ms (SD = 39.78 ms). Therefore, compared to the TD children, the FR children had a significantly later p-MMR latency, suggesting, plausibly, more effortful discrimination of the two non-words. For the single-speaker condition, group showed no significant effect, F(1, 66) = 3.64, *p* = 0.061. Thus, no difference was found between the groups for the MMR peak latency in the single condition.

To examine how the TD and FR groups differed in their mismatch responses in the two conditions, a mixed-effect ANOVA was conducted, with group being the between-subject variable, and stimulus type (standard, deviant), conditions (single-, multiple-speaker), lateralization (left, middle, right), and location (frontal or central) being the within-subject variables. Crucially, type (standard or deviant) showed a significant main effect, F(1, 66) = 10.24, *p* = 0.002, partial η^2^ = 0.13. Furthermore, condition showed a significant main effect, F(1, 66) = 4.11, *p* = 0.047, partial η^2^ = 0.06, indicating larger ERP amplitude in the multiple-speaker condition. Location showed a significant main effect, F(1, 66) = 25.82, *p* < 0.001, partial η^2^ = 0.28, and lateralization showed a significant main effect F(2, 132) = 24.63, *p* < 0.001, partial η^2^ = 0.27, indicating larger ERP amplitude at frontal than at central electrodes, and larger ERP amplitude at left than right electrodes. The interaction between type and location was significant, F(1, 66) = 39.95, *p* < 0.001, partial η^2^ = 0.38, and so was the interaction between location and lateralization, F(2, 132) = 3.79, *p* = 0.03, partial η^2^ = 0.05. Together with Figure 2, it can be seen that the p-MMR was more evident at frontal than central electrodes. The interaction between type and lateralization was significant, F(2, 132) = 11.61, *p* < 0.001, partial η^2^ = 0.15, indicating left-lateralized p-MMR. Importantly, the interaction between type and group was not significant, F(1, 66) = 0.63, *p* = 0.43, nor was the interaction between lateralization and group, F(2, 132) = 0.34, *p* = 0.72; the three way interaction between condition, type, and group was also not significant, F(1, 66) = 1.25, *p* = 0.27.

As can be seen in Figure 2, these results show that the ERPs to the deviant were significantly more positive than those to the standard, indicating an overall p-MMR for both the TDs and the FRs across the two conditions, and the p-MMR was more evident at the frontal than the central electrodes. With regard to condition, for both the TDs and the FRs, the overall ERPs in the multiple-speaker condition were larger than those in the single-speaker condition. Importantly, there was no evidence that the TDs and FRs differed in terms of p-MMR amplitude or lateralization, and the p-MMRs of both groups were comparable in terms of amplitude across the two conditions.

When the more conservative non-parametric cluster analysis was performed, no significant cluster was found for any group. As the 32-channel cap did not allow a dense spatial sampling, it was possible that the sparse distribution of electrodes may have rendered the cluster analysis non-significant.

## 3. General Discussion

In the current study, 20-month-old typically developing children and children at familial risk of dyslexia were tested on their neural categorization of the acoustically similar native vowel contrast /i/ and /I/ realized in two nonwords [ɣIp] and [ɣip]. Overall, both the TD and FR children showed a significant p-MMR, and the effect of the condition was not significant. There was no evidence for different scalp distributions for the MMRs among the TDs and FRs. The significant interaction between type and location indicates a more left-lateralized p-MMR for both conditions and for both groups. These results indicate that both groups were able to neurally discriminate the two non-words, regardless of whether speaker variability was introduced. Importantly, there was no evidence that p-MMR was attenuated among the FRs as compared to the TDs. Nevertheless, the MMR peak latency of the FR children was significantly later than that of the TD children. These findings suggest that although FR children were able to categorically discriminate the two non-words, their discrimination might be more effortful than the TDs.

Unlike other studies (e.g., [33,65,76]) that found less categorical phoneme perception in FRs, the current study found comparable MMR amplitude between the FR and the TD children in the multiple-speaker condition. Seeing that [76] and [79] made use of synthesized monosyllables while the stimuli of the current study were natural productions that were minimally manipulated, it might be that FR children were hampered when the stimuli were less rather than more speech-like. The temporal sampling framework for developmental dyslexia hypothesizes that segmental-level representations might be too specific, while syllable-level representations are deficient in dyslexia, due to impaired low- but not high-frequency neural modulation. The current study did not distinguish between segmental and syllabic difference, and it would be informative for future studies to investigate whether syllable-level variability (e.g., variability in amplitude modulation) is more problematic than segmental-level (e.g., formant structures) variability for dyslexics. So far, MMN studies with FR children are still limited, and different studies testing children of different ages with different stimuli and more evidence is needed to understand when and under what circumstances MMN differs between TD and FR children.

Nevertheless, the FR children had a significantly later p-MMR peak latency compared to the TD children in the multiple-speaker condition. Seeing that MMN peak latency has been found to increase with a decrease in deviation magnitude [47], it seems that when presented with the same stimuli, the difference between the standards and the deviants might have been perceived as smaller among the FRs than among the TDs, suggesting more effortful neural discrimination for the FRs [63,64,65]. Previous research has shown that dyslexic children can be less capable of learning statistical contingencies between visual and phonological cues [108,109]; whether such difficulties may relate to dyslexic children’s characteristics of neural categorization needs further investigation.

For both groups and for both conditions, all the mismatch responses were still positive. Therefore, although the children were able to normalize speaker variability and discriminate the vowels categorically, at this age the mismatch response was still not adult-like, which was consistent with previous studies testing neural discrimination of non-salient native contrasts [60,110]. Therefore, for both the TDs and the FRs, maturation of mismatch response (i.e., a shift to adult MMN) is expected to continue, yet when exactly such a shift will occur should be investigated in future studies.

In the current study, vocabulary level was controlled between the FRs and TDs. The lexical restructuring model (LRM, [84]) hypothesizes that words are represented holistically at the initial stage of word learning, and, as infants’ vocabulary expands, to deal with increases in neighborhood density, infants are forced to refine phonetic representations. In parallel with vocabulary expansion, the similar-sounding phonetic categories will reorganize and become represented in a more adult-like way [84,111,112]. According to LRM, at the initial stage of word learning, phonological categorization is expected to be more evident for children who have a large vocabulary. Given that previous studies showed FRs to have smaller vocabularies than TDs at 20 months [86], it is plausible that FRs might be less capable of phonological categorization than TDs as a result of small vocabularies. The current study, however, matched the TD and FR children on vocabulary, yet we still observed a group difference in the single-speaker condition. Therefore, it seems likely that the FRs, even when comparable with TDs in terms of vocabulary development, still had a different mode of neural discrimination of phonetic differences.

It should be acknowledged that the FR children tested in the current study only had elevated risk for becoming DD and not all of them will become DD in the end [32,113]. For now, it is difficult to ascertain whether the FRs who would become DD differed in phonetic and phonological discrimination from those who would not. Once their reading status becomes available, it will be worth the effort to compare these two subgroups of FRs on their MMRs to investigate whether neural signatures underlying phonetic and phonological discrimination can be considered reliable precursors of DD.

## 4. Conclusions

To conclude, the FRs’ p-MMR magnitude was comparable to that of the TDs when discriminating a native non-salient vowel contrast regardless of whether inter-speaker variability was present, and phoneme categorization difficulty does not qualify as a labeling feature for FRs.

## Figures and Tables

**Figure 1 brainsci-12-00412-f001:**
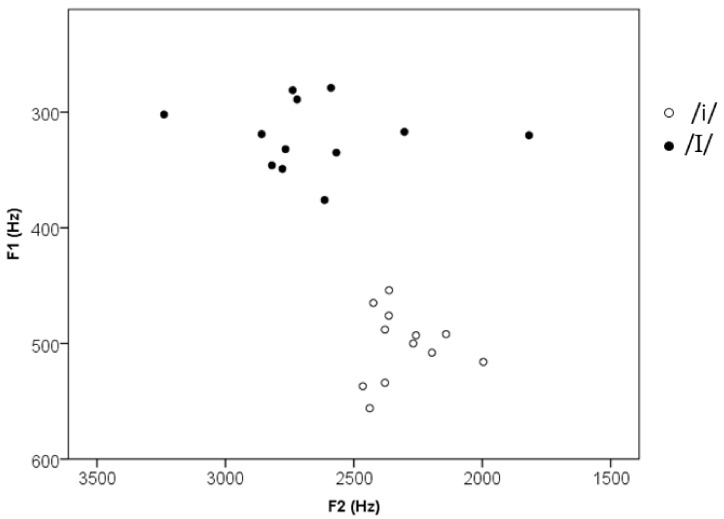
Scatterplot of the F1 and F2 values of the vowels in the stimuli.

**Figure 2 brainsci-12-00412-f002:**
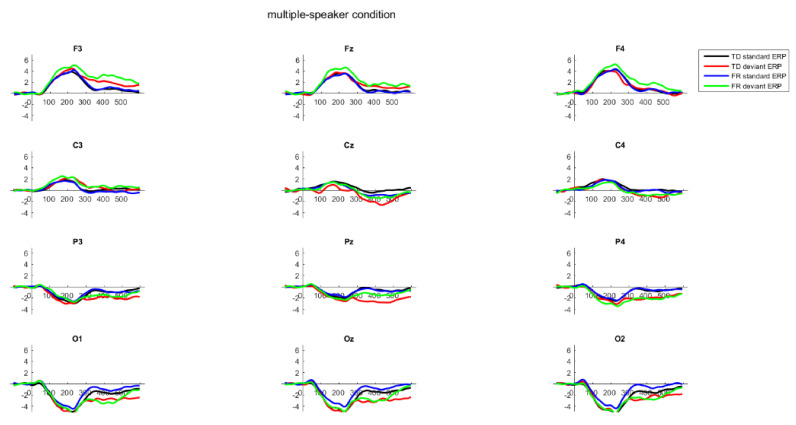
Standard ERPs, deviant ERPs, and difference waves of the typically developing TD children and children at familial risk of dyslexia (FR) in the multiple-speaker and single-speaker condition. The bars in the difference wave graphs indicate the time windows where the mismatch responses (MMRs) MMRs were significant, with the bars’ colors corresponding to group membership. Bars above the ERP waves indicate positive MMR, and bars below the ERP waves indicate negative MMR.

**Figure 3 brainsci-12-00412-f003:**
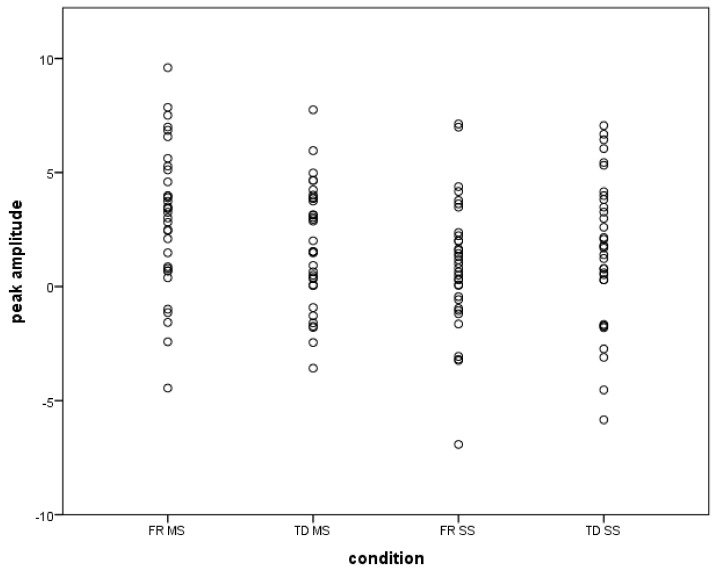
MMR peak amplitude at F3 of individual participants in the multiple- and single-speaker conditions. FR = familial risk group, TD = typically developing group, MS = multiple-speaker condition, and SS = single-speaker condition. Each circle represents an individual participant.

**Table 1 brainsci-12-00412-t001:** Characteristics of the typically developing (TD) andchildren at familial risk of dyslexia (FR). CDI = Dutch version McArthur Communicative Development Inventory.

	Sample Size	Age (SD) in Days at the Experiment	Gender	CDI Productive Score	CDI Comprehensive Score	Days (SD) between CDI and Experiment
TD	35	601 (13)	16 girls	97 (90)	297(131)	7 (11)
FR	35	604 (20)	16 girls	98 (84)	296 (107)	6 (12)

**Table 2 brainsci-12-00412-t002:** Mean (SD, range) f0, F1, F2, F3 and duration of vowels /I/ in [γIp] and /i/ in [γip], and results of the *t*-tests for comparison of each of these measurements.

	Multiple-Speaker Condition		
	**/i/ in *giep***	**/I/ in *gip***	**t (11)**
f0 max (Hz)	269 (34, 219–321)	243 (60, 126–321)	0.89, *p* = 0.39
f0 min (Hz)	226 (32, 184–272)	216 (55, 120–294)	0.35, *p* = 0.56
F1 (Hz)	320 (29, 279–376)	502 (30, 454–556)	**13.72, *p* < 0.001**
F2 (Hz)	2651 (342, 1818, 3239)	2636 (139, 1996–2465)	**3.75, *p* = 0.003**
F3 (Hz)	3189 (231, 2724–3624)	2896 (191, 2601–3193)	**3.19, *p* = 0.009**
Vowel duration (ms)	91 (16, 66–122)	88 (11, 70–113)	0.87, *p* = 0.41
Initial consonant duration (ms)	87 (2, 83–89)	88 (2, 86–91)	1.76, *p* = 0.12
Word duration (ms)	346 (9, 331–361)	342 (10, 323–357)	1.04, *p* = 0.32

**Table 3 brainsci-12-00412-t003:** TD and FR mean (SD) (percentage of total trials) accepted trials in the multiple-speaker (MS) and single-speaker (SS) condition.

	MS Standard	MS Deviant	SS Standard	SS Deviant
TD	279 (34) (78%)	93 (12) (78%)	266 (38) (74%)	89 (13) (74%)
FR	286 (64) (79%)	96 (21) (80%)	275 (30) (76%)	93 (12) (78%)

## Data Availability

The data presented in this study are available on request from the corresponding author. The data are not publicly available due to privacy reasons.

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
