# Peer review of "Later but Not Weaker: Neural Categorization of Native Vowels of Children at Familial Risk of Dyslexia"

_brainsci, 2022, doi:10.3390/brainsci12030412_

Round 1

Reviewer 1 Report

I find that the present version of the manuscript is an improved version of the original one. I thank the author for the important effort made in order to respond to most of the points raised by the reviewers, and adapt the manuscript accordingly. Therefore, I can recommend this document for publication without further suggestions.

Author Response

I thank the reviewer for the very helpful comments, which has greatly improved the paper. 

Reviewer 2 Report

The manuscript at hand is a study I reviewed earlier. The author tried to respond to all my previous comments. 

The author interprets group-differences in the single-speaker condition. This is not the case. Even though t-tests showed effects in the TD group, but not in the FR group, the specific difference between groups cannot be proven statistically. Thus, no group difference can be interpreted, which changes the focus of the manuscript immensely.

The results (still) include effects not relevant to the research question. The author is interested in effects involving the factor stimulus type (standard, deviant). This factor would be the main factor of interest, because it tells the author whether there was a significant MMR. All other effects that do not involve the factor stimulus type cannot be interpreted in terms of the MMR and are therefore not relevant to the research question. The following effects can actually be interpreted:

Main effect of stimulus-type
Interaction between stimulus-type and electrode.

No other effect can be used for interpretation and discussion of the MMR. Thus, the author did not find any effects supporting differences between groups (at least not for amplitude), differences between conditions (single vs. multiple-speaker) and/or differences between groups depending on condition. I am very suspicious whether these data can really be used for the conclusion that TDs and FRs process phonetic differences differently in the single-speaker condition and/or whether there are differences between conditions at all. This has not been confirmed by the analyses.

Further, the author described in the reviewer response that he*she also performed analyses with central electrodes. This should then be described in the main manuscript. Otherwise, it can also not be concluded that the MMR had a frontal distribution. Further, as suggested by another reviewer, more than 3 (6) electrodes should be included in analyses (maybe even all). The author could also use regions of interest, if cluster-based permutation does not work out. That cluster-based permutation tests did not work out, might also be a problem for the present manuscript, but I sympathize with the author as infant data are often difficult in terms of cluster-based permutation tests.

Additionally, I don’t think that discussion on scalp distribution should be toned down, but it should be excluded. There is no evidence in the data for differences in scalp distribution at all.

Further comments:

  • I strongly suggest re-analyzing the data without the two participants providing little deviant trials.
  • Page 8, line 361: group is not a within-subject variable.

Author Response

The manuscript at hand is a study I reviewed earlier. The author tried to respond to all my previous comments. 

The author interprets group-differences in the single-speaker condition. This is not the case. Even though t-tests showed effects in the TD group, but not in the FR group, the specific difference between groups cannot be proven statistically. Thus, no group difference can be interpreted, which changes the focus of the manuscript immensely.

The results (still) include effects not relevant to the research question. The author is interested in effects involving the factor stimulus type (standard, deviant). This factor would be the main factor of interest, because it tells the author whether there was a significant MMR. All other effects that do not involve the factor stimulus type cannot be interpreted in terms of the MMR and are therefore not relevant to the research question. The following effects can actually be interpreted:

Main effect of stimulus-type
Interaction between stimulus-type and electrode.

No other effect can be used for interpretation and discussion of the MMR. Thus, the author did not find any effects supporting differences between groups (at least not for amplitude), differences between conditions (single vs. multiple-speaker) and/or differences between groups depending on condition. I am very suspicious whether these data can really be used for the conclusion that TDs and FRs process phonetic differences differently in the single-speaker condition and/or whether there are differences between conditions at all. This has not been confirmed by the analyses.

Response: I removed the two participants with less than 30 accepted deviant trials as suggested by the reviewer, and all the analysis has been redone with the remaining 34 TD and 34 FR children.

I also included the central electrodes (i.e., C3, Cz, C4) as the reviewer suggested. The revised analysis found significant main effect of condition (single- or multiple-speaker), type (standard or deviant), location (frontal or central), and lateralization (left, middle, right). As the reviewer suggested, only significant interactions involving type were interpreted. The interaction between type and lateralization was significant, confirming a left lateralized p-MMR for both groups and both conditions. A further interaction between type and location confirms the frontal distribution of the p-MMR. None of the other interactions turned out to be significant.

            After removing the two participants, the revised analyses found significant p-MMRs in both conditions for both groups. Therefore, indeed, as the reviewer pointed out, there was no evidence that the two groups differed in terms of the presence of the p-MMR or lateralization. The results part has been rewritten, and the discussion on group difference and lateralization difference has been removed.

        The Figure 2 and Figure 3 in the results part have been revised by removing the two participants. Table 4 and Table 5 presenting the windows where the standard and deviant ERPs differed significantly and the mean ERP amplitudes have been moved to supplementary materials.

Please see the revised analysis:

“To examine how the TD and FR groups differ in their mismatch responses in the two conditions, a mixed-effect ANOVA was conducted with group being the between-subject variable, and stimulus type (standard, deviant), conditions (single-, multiple-speaker), lateralization (left, middle, right), and location (frontal or central) being the within-subject variables. Condition showed a significant main effect, F(1, 66) = 4.11, p = .047, partial η2 = 0.06, type (standard or deviant) showed a significant main effect, F(1, 66) = 10.24, p = .002, partial η2 = 0.13, location showed a significant main effect, F(1, 66) = 25.82, p < .001, partial η2 = 0.28, and lateralization showed a significant main effect F(2, 132) = 24.63, p < .001, partial η2 = .27, with the ERP amplitude at the left electrodes being more positive than that of the middle and right electrodes. The interaction between type and location was significant, F(1, 66) = 39.95, p< .001, partial η2 = 0.38, and so was the interaction between location and lateralization, F(2, 132) = 3.79, p = .03, partial η2 = 0.05. Together with Figure 2, it can be seen that the p-MMR was more evident at frontal than central electrodes. The interaction between type and lateralization was significant, F(2, 132) = 11.61, p < .001, partial η2 = 0.15, indicating left-lateralized p-MMR. Importantly, the interaction between type and group was not significant, F(1, 66) = 0.63, p = .43, and neither was the interaction between lateralization and group, F(2, 132) = 0.34, p = .72; the three way interaction between condition, type, and group was also not significant, F(1, 66) = 1.25, p = .27.” 

In the Discussion part, p23, line 7-10:

"There was no evidence for different scalp distribution of the MMRs among the TDs and FRs. The significant interaction between type and location indicates a more left-lateralized p-MMRs for both conditions and for both groups. "

Further, the author described in the reviewer response that he*she also performed analyses with central electrodes. This should then be described in the main manuscript. Otherwise, it can also not be concluded that the MMR had a frontal distribution. Further, as suggested by another reviewer, more than 3 (6) electrodes should be included in analyses (maybe even all). The author could also use regions of interest, if cluster-based permutation does not work out. That cluster-based permutation tests did not work out, might also be a problem for the present manuscript, but I sympathize with the author as infant data are often difficult in terms of cluster-based permutation tests.

Response: Please see my response to the previous point.

Additionally, I don’t think that discussion on scalp distribution should be toned down, but it should be excluded. There is no evidence in the data for differences in scalp distribution at all.

Response: the discussion on scalp distribution has been removed. 

Further comments:

  • I strongly suggest re-analyzing the data without the two participants providing little deviant trials.

Response: the two participants were excluded in the revised manuscript.

  • Page 8, line 361: group is not a within-subject variable.

Response: the text has been revised to “To examine how the TD and FR groups differ in their mismatch responses in the two conditions, a mixed-effect ANOVA was conducted with group being the between-subject variable, and stimulus type (standard, deviant), conditions (single-, multiple-speaker), lateralization (left, middle, right), and location (frontal or central) being the within-subject variables.”.

Reviewer 3 Report

Neural Categorization of Native Vowels of Children at Familial Risk of Dyslexia by Ao Chen

The author compares categorization of vowels of typically developing and children with familiarity of dyslexia. They did it in preschool 35 infants, 20 months old. Typically, they opened their questionnaire two nonwords “giep” [É£ip] and “gip” [É£Ip] that were con solely by the vowel In the multiple-speaker condition, nonwords were produced by twelve speakers while in the single-speaker condition, only one produced them

They looked at behavioural and EEG (positive mismatch response)

Results

In the multiple-speaker condition

  • positive mismatch response (p-MMR) elicited in both groups
  • p-MMRs were of comparable amplitude across the two groups
  • FR children had a later p-MMR peak than the TD children

single-speaker cathegorization condition

              TDs but not FRs showed a significant p-MMR

The experiment and the results seems sound, and the results are quite interesting. The main point I have is that it is difficult for me to associate multple spekers to specific phonetic modulation (which is, if I understand well , the argument of the author).

  • The authors have a large and interesting question in the introduction on phonological awareness and on phonology, but say that they test the melody phonetic by implicating many different voices: Can the author clearly explain what is in his experiment,, phonetic, melodic and phonology, in order for reader to understand why melody and phonology should be tested her ? Particularly, the voice of another speakers has to do with phonetic but also with Melody prosody, and also phonological differences.
  •  
  • Maybe I missed it but I did not see control condition, such as non verbal stimuli : Please explain why; Do the author have data on non verbal stimuli
  • Can the authors discuss the MMR meaning a little more of p-MMR peak delay in FR children. Could that be in phase with difficulty of phonological visual integration  (see for example MW jones 2018)
  •  

Author Response

The author compares categorization of vowels of typically developing and children with familiarity of dyslexia. They did it in preschool 35 infants, 20 months old. Typically, they opened their questionnaire two nonwords “giep” [É£ip] and “gip” [É£Ip] that were con solely by the vowel In the multiple-speaker condition, nonwords were produced by twelve speakers while in the single-speaker condition, only one produced them

They looked at behavioural and EEG (positive mismatch response)

Results

In the multiple-speaker condition

  • positive mismatch response (p-MMR) elicited in both groups
  • p-MMRs were of comparable amplitude across the two groups
  • FR children had a later p-MMR peak than the TD children

single-speaker cathegorization condition

              TDs but not FRs showed a significant p-MMR

The experiment and the results seems sound, and the results are quite interesting. The main point I have is that it is difficult for me to associate multple spekers to specific phonetic modulation (which is, if I understand well , the argument of the author).

  • The authors have a large and interesting question in the introduction on phonological awareness and on phonology, but say that they test the melody phonetic by implicating many different voices: Can the author clearly explain what is in his experiment,, phonetic, melodic and phonology, in order for reader to understand why melody and phonology should be tested her ? Particularly, the voice of another speakers has to do with phonetic but also with Melody prosody, and also phonological differences.

Response: The study did not aim to test melody. The focus of the current study was to test whether the FR children were less able to process variability when categorizing native vowels. Therefore, the design was to contrast two conditions that differed in terms of whether variability was present, not to contrast different types of variability (e.g., phonetic or melodic). In the multiple-speaker condition, different speakers were used to introduce variability, which contrasts with absence of variability in the single-speaker condition. In the multiple-speaker condition, only if the children were able to categorize the variable voices to one single nonword, a mismatch response could be elicited. Thus, elicitation of MMR in the multiple-speaker condition would require phonological processing of the nonwords.

      I added explanation of the design in the introduction, on page 9, line 10-12:

In particular, to understand whether the two groups differed in phonetic discrimination or phonological categorization, two experimental conditions were designed. In the single-speaker condition, one token of [É£Ip] and one token of [É£ip], both produced by the same speaker, were used as stimuli, while in the multiple-speaker condition, twelve tokens of each nonword, each produced by a different speaker were used as stimuli. Since the stimuli were obtained by manipulating natural production, in the multiple-speaker condition, the tokens differed in melodic contour (f0), voices, and phonetic implementation of the vowels. In essence, speaker normalization and classical categorization experiments require the same skill, namely that listeners should attend to acoustical differences that signal phonologically relevant speech contrasts while disregard signals irrelevant for linguistic content. Indeed, findings on speaker normalization are similar to the classical categorical perception experiments, namely that children with dyslexia were less able to normalize speaker variability [79].

                  On page 3, line 1-2 from the bottom:

“In categorical perception experiments, listeners are presented with stimuli that gradually change from one phonological category to another in a step-wise fashion, such from pea to bee. Typical listeners perceive such gradual changes as having distinct categories (i.e., identification shifts from pea to bee consistently at one specific step), and they discriminate the between-category stimuli better than the within-category ones. The essence of categorical perception is that listeners disregard those lexically irrelevant acoustical differences while attend the differences of same magnitude if these change word meaning. Categorical perception is essential for efficient speech processing since variability abounds in human speech. The same phoneme or word can drastically differ acoustically due to speakers’ anatomical structures, speaking rate, and phonetic context, etc. [20]–[22].”

  • Maybe I missed it but I did not see control condition, such as non verbal stimuli : Please explain why; Do the author have data on non verbal stimuli

Response: the focus of the current study was test speech processing of the FR and TD children, and to contrast acoustic/phonetic processing of speech sounds (i.e., in the single-speaker condition where no variability was introduced) and phonological categorization of speech sounds (i.e., in the multiple-speaker condition where variability was introduced). Therefore, I didn’t test the children on non-speech stimuli.

  • Can the authors discuss the MMR meaning a little more of p-MMR peak delay in FR children. Could that be in phase with difficulty of phonological visual integration  (see for example MW jones 2018)

Response: In the discussion of the p-MMR peak latency, I added the discussion on phonological visual integration. Please see on page 24, line 12-15:

“Seeing that MMN peak latency has been found to increase with decrease in deviation magnitude [46], it seems that when presented with the same stimuli, the difference between the standards and the deviants might have been perceived to be smaller among the FRs than among the TDs, suggesting more effortful neural discrimination of the FRs. Previous research has shown that dyslexic children can be less capable of learning statistical contingencies between visual and phonological cues [105], and whether such difficulties may relate to the dyslexic children’s characteristics of neural categorization needs further investigation.

Round 2

Reviewer 2 Report

The manuscript at hand is a manuscript I reviewed earlier. The author has further improved the manuscript by responding to the reviewer comments. However, I think there are still some points that should be addressed before this manuscript can be accepted for publication.

First and most importantly, the focus of the manuscript has changed due to previous reviewer comments. This, however, should also include introduction and more importantly discussion. The author should put more emphasize on discussing and introducing latency differences between groups. What could latency differences between groups mean? Why are there latency, but no amplitude differences between groups. Why were there latency differences in the multiple-, but not single-speaker condition?

Second, there are still paragraphs in the manuscript that suggest amplitude differences between groups. For example:

  • In the abstract
  • In the discussion on page 25, line 18 and following.

Third, I noticed that degrees of freedom differ between ANOVAs on latency and ANOVAs on amplitude. Why is this the case? Did the author perform ANOVAs with different samples (sizes)?

Fourth, I still think it would be better to move all effects not involving the factor type to the supplements. Otherwise, the results section is overwhelming and difficult to follow.

Author Response

The manuscript at hand is a manuscript I reviewed earlier. The author has further improved the manuscript by responding to the reviewer comments. However, I think there are still some points that should be addressed before this manuscript can be accepted for publication.

First and most importantly, the focus of the manuscript has changed due to previous reviewer comments. This, however, should also include introduction and more importantly discussion. The author should put more emphasize on discussing and introducing latency differences between groups. What could latency differences between groups mean? Why are there latency, but no amplitude differences between groups. Why were there latency differences in the multiple-, but not single-speaker condition.

Response: discussion of the findings on MMR latency difference has been added, please see:

In Introduction on page 3, line 114-115:

Atypical MMN has been commonly reported for people with DD [19], [61], and a recent meta-analysis has shown that irrespective of age, speech elicited MMN is aberrant among people with DD [62]. DD and FR children has been reported to exhibit later MMN peak latencies, possibly reflecting more effortful processing of speech sounds [63]–[65].

In Discussion on p11-p12, line 457-460

These results indicate that both groups were able to neurally discriminate the two non-words, regardless whether speaker variability was introduced. Importantly, there was no evidence that the p-MMR was attenuated among the FRs as compared to the TDs. Nevertheless, the MMR peak latency of the FR children was significantly later than that of the TD children. These findings suggest that although FR children were able to categorically discriminate the two non-words, their discrimination might be more effortful than the TDs.

In discussion on p12 line 478-481:

Nevertheless, the FR children had a significantly later p-MMR peak latency compared to the TD children in the multiple-speaker condition. Seeing that MMN peak latency has been found to increase with decrease in deviation magnitude [47], it seems that when presented with the same stimuli, the difference between the standards and the deviants might have been perceived to be smaller among the FRs than among the TDs, suggesting more effortful neural discrimination of the FRs [63]–[65].

Second, there are still paragraphs in the manuscript that suggest amplitude differences between groups. For example:

  • In the abstract
  • In the discussion on page 25, line 18 and following.

Response: the abstract has been revised, and the paragraph on amplitude in the discussion part has been removed.

Abstract:

Abstract: Although allophonic speech processing has been hypothesized to be a contributing factor for developmental dyslexia, experimental evidence is limited and inconsistent. The current study compared categorization of native similar sounding vowels of typically developing (TD) children and children at familial risk (FR) of dyslexia. EEG response was collected in a non-attentive passive oddball paradigm from 35 TD and 35 FR Dutch 20-month-old infants who were matched on vocabulary. The children were presented with two nonwords “giep [É£ip] and “gip” [É£Ip] that were contrasted solely by the vowel. In the multiple-speaker condition, both nonwords were produced by twelve different speakers while in the single-speaker condition, one single token of each word was used as stimuli. For both conditions and for both groups, infant positive mismatch response (p-MMR) was elicited, and the p-MMR amplitude was comparable between the two groups. although the FR children had a later p-MMR peak than the TD children in the multiple-speaker condition. These findings indicate that the FR children are able to categorize speech sounds, but may be in a more effort way than the TDs.

Removed paragraph in Discussion:

The findings of the current study do not seem to guarantee phoneme categorization difficulty as a labeling feature for the FRs, since their MMR in the multiple-speaker condition was comparable with that of the TDs. Interestingly, it was in the single-speaker condition that the FRs seemed to discriminate the two vowels less well than the TDs. Thus, the phonetic discrimination and phonological categorization does not seem to develop in parallel among the FRs, and their establishment of phoneme representation may have a different developmental trajectory compared to the TDs. Therefore, whether aberrant phonetic discrimination underlies phonological categorization among the FRs and dyslexics, and how these two relate to dyslexics’ deeply rooted phonological awareness difficulty needs further investigation. It is important for future studies to capture and compare phoneme learning of the TDs and FRs in a progressive manner with longitudinal designs.

Third, I noticed that degrees of freedom differ between ANOVAs on latency and ANOVAs on amplitude. Why is this the case? Did the author perform ANOVAs with different samples (sizes)?

Response: My apologies for the typo in the latency analysis. Reports of the ANOVA results have been revised, the DF should be (1,66). Please see:

For the multiple speaker condition, F (1, 66) = 36.50, p < .001, partial η2 = .36, For the single-speaker condition, group showed no significant effect, F(1, 66) = 3.64, p = .061.

Fourth, I still think it would be better to move all effects not involving the factor type to the supplements. Otherwise, the results section is overwhelming and difficult to follow.

Response: The main effected of condition, lateralization, and location were reported to demonstrate the scalp distribution of the ERPs. “Crucially” was added to highlight the effect of type. Only interactions with type involved were reported. Please see:

a mixed-effect ANOVA was conducted with group being the between-subject variable, and stimulus type (standard, deviant), conditions (single-, multiple-speaker), lateralization (left, middle, right), and location (frontal or central) being the within-subject variables. Crucially, type (standard or deviant) showed a significant main effect, F(1, 66) = 10.24, p = .002, partial η2 = 0.13. Besides, condition showed a significant main effect, F(1, 66) = 4.11, p = .047, partial η2 = 0.06, indicating larger ERP amplitude in the multiple-speaker condition. Location showed a significant main effect, F(1, 66) = 25.82, p < .001, partial η2 = 0.28, and lateralization showed a significant main effect F(2, 132) = 24.63, p < .001, partial η2 = .27, indicating larger ERP amplitude at frontal than central electrodes, and larger ERP amplitude at left than right electrodes.

Reviewer 3 Report

ok for me with these important explanations

Author Response

Dear Reviewer,

Many thanks for the very helpful comments. 

This manuscript is a resubmission of an earlier submission. The following is a list of the peer review reports and author responses from that submission.

Round 1

Reviewer 1 Report

I find this study to be very interesting. The introduction is very concise, informative and well written. The authors summarize thoroughly the background on developmental dyslexia, categorical perception, MMN variability, etc. A question under debate is whether TF and FR infants present differences in phonetic discrimination or phonological categorization. Hence, the authors use the MMN paradigm to shed light on this issue by means of including 2 stimuli conditions: the single speaker and the multiple speaker condition. 

The rationale of participant selection and stimuli creation (vowels) are clearly described.  

However, I have important concerns regarding several methodological issues.  

For example, regarding the order of blocks, the authors explain hat “as the focus of the current study was speaker normalization, the multiple-speaker block always preceded the single-speaker condition”. I do not follow the logic of this argument. Please, explain (lines 315-320). 

Regarding the visual display of the data there are several problems I outline bellow: 

  • First, the authors present the results in 12 electrodes. However, the statistical analysis is carried out only on 3 electrodes. Please correct. 
  • Second, they choose F3, Fz and F4 based on Noordenbos et al 2012 study. However, this article does not include any info on EEG data. Hence, the authors shall include an appropriate justification of the electrode selection for the MMR/MMN analysis.
  • In this line, limiting the analysis to just 3 of 32 channels may provide a rather poor picture of the processes underneath. Nowadays there are methods for more extensive and objective statistical analysis of multi-channel ERP data which are available in most software packages (e.g. cluster-based permutation tests).  
  • Regarding the latency analysis, the authors state that MMR peak amplitudes were calculated as the mean amplitude in the 40 ms (…) surrounding the grand average peak. Is this correct? To me, it is not, as they do not use the information already calculated –the individual peak latency- in order to obtain the individual peak amplitude. This is: as it reads now, the individual peak amplitude does not refer to individual peak latency. Please, check. 
  • I suggest the authors to present the information in table 4 in a graphical way. There is (free) software available to create figures that show the time-course of the differences in each of the electrodes under analysis. This shall facilitate readers’ comprehension.  
  • - Regarding the waves and the figures, the authors shall present a figure with the difference waves for each condition and group. This would help the authors and the readers understand the results better. 
  • Regarding table 3: the authors shall present the values in %. Besides, statistical analysis –ANOVAs- shall be applied in order to discard significant differences with respect to the number of items included for the analysis. Otherwise, in case there are differences, the authors shall explain why that is the case.  
  • Finally, in the Conclusion section, the authors state that scalp distribution of the MMR was different across the groups. However, the electrode sample was chosen to carry out the statistical analysis was rather small (3 electrodes). I suggest the authors to proceed with a larger number of electrodes (maybe all?) in order to have an expanded picture of the present results.  

Reviewer 2 Report

Summary

This manuscript reports results from an experimental study with infants, which were split in two groups, typically developing and with familial risk of dyslexia. The study focused on speech perception and processing of native similar sounding vowels in multiple-speaker and single-speaker conditions.

Thank you for the opportunity to review this manuscript. I enjoyed reading this manuscript and I think it has considerable potential to contribute to the literature in several ways. It can add further evidence to the literature on better understanding the profile that children with familial risk of dyslexia show compared to TD children.

Introduction

Very clear and detailed account of a wide range of studies on the differences that dyslexic children present. I think the very comprehensive review of the studies on phonological awareness and speech perception differences really help build the rationale for this study. Highlighting studies, like Richardson et al (2003), further strengthens the rationale for this study in terms of the differences that children with familial risk of dyslexia show compared to TD children. The gap in further research on MMN in children with familial risk of dyslexia has also been emphasised well which further demonstrates the importance of this study for the field of dyslexia research.

Method

In the participants section, please clarify for the reader the rationale behind the decision to have unequal numbers in each experimental group (53 TDs vs 35 FRs). Could you also clarify how from 36 FRs who initially participated, the final number is 35 after four were excluded for various reasons. This adds up to 32 for me, so that needs editing for clarity.

Please also clarify the rationale behind the choice of tests for the FR parents. I understand and support the inclusion of the two reading tests, but it is not clear why the WAIS comprehension tasks. Was that the Verbal Comprehension Index which includes the Vocabulary, Similarities and Comprehension subtests or did you only do the Comprehension subtest? Dyslexia doesn’t tend to affect listening comprehension, which the WAIS task mainly assess, so it is essential that the rationale for this assessment is further emphasised.

It is also important in my opinion to include some further details about these tasks in the same way that there is description for the reading tasks, so that it is clear what participants were required to do in each task.

Could you also please include which version of the WAIS you used? I would imagine that it might be WAIS-IV, but please clarify this for the reader.

There is a clear description and outline of the process used to generate the stimuli for the speech processing task. The chosen words and non-words seem relevant, and the researcher has clearly considered carefully which CVC words to use to avoid confusion, while also taking into account the specific characteristics of Dutch grammar and syntax.

A very detailed account of the procedure followed was included, thank you for that.

Results

Very clear and detailed presentation of the statistical analyses and I think the inclusion of the ERP results in figures helps highlight the key findings for the experimental groups and conditions.

Discussion

The author has provided a comprehensive description of the many findings and an explanation of their findings in the light of previous literature. The results provide an interesting contribution to the existing literature in terms of the characteristics that children with familial risk of dyslexia present in terms of speech perception and processing. The limitations of the study have clearly been considered and the need for further research on this topic has been emphasised. The conclusion highlights well the key finding around the differences in phonological and phonetic processing mechanisms for typically developing children and children with familial risk of dyslexia.

Referencing

As far as I know, citations need to be presented in numbers within brackets (e.g. [1]), which did not seem to be the case in this manuscript. I would therefore recommend significant changes to the way that in-text citations and the reference list are presented in order to be in line with the journal’s guidelines.

Reviewer 3 Report

The manuscript at hand is on a study investigating differences between 20-month-olds with (FR) and without (TD) familial risk of dyslexia on phoneme categorization vs. phoneme perception. To test this, the author investigated the Mismatch Negativity (MMN) in the event-related potential (ERP) in two conditions. The first condition, referred to as the single-speaker condition, used two nonwords that contrasted in vowel, and which were spoken by one speaker. The multiple-speaker condition, in contrast, had 12 different speakers, but used the same two nonwords that contrasted in the vowel. The author argues that the single-speaker condition can be used for investigating phoneme perception, while the multiple-speaker condition can be used to test phoneme categorization as listeners need to extract the relevant vowel categories for discrimination, while "ignoring" the characteristics of the different speakers, if I understood correctly.

The author argues that the results show that phoneme processing differed between TD and FR infants, indicated by no Mismatch Response (MMR) in FR infants, but significant MMR in TD infants in the single-speaker condition. Further, the multiple-speaker condition did not show significant differences between groups, which can, according to the author, be interpreted such that phoneme categorization does not differ between the groups.

While this study is generally of interest, I have some problems with analysis (preprocessing and statistical analysis), which I will describe below. I am not sure whether the results that the author describes are statistically valid. Further, the manuscript needs a thorough spell and grammar check. 

In the following, I will describe my concerns and more minor comments on the manuscript.

Abstract, page 1, line 20: I think that a word is missing - it should be: may be processing, instead of may processing.
Independently, the sentence is not entirely informative. What does it exactly mean that processing is different, but categorization works? I think it would be helpful to, already in the abstract, explicitly state which experimental condition tests categorization and which tests processing.

Page 2, line 46: changed should be change. The manuscript should have a thorough spell and grammar check by an English native speaker.

Page 2, line 59 and following: The author describes that findings show different results concerning the ability to form and process phoneme categories as well as concerning phoneme perception abilities. The paragraph was somehow difficult to follow because perception was used for phoneme perception and categorical perception of phonemes. Specifically, when reading the last sentence stating that it remains unclear whether atypical categorical perception can be considered as a labeling feature for DD, or whether it only applies to a subgroup, I got confused, because in the sentence before, the author says that only subgroups of children with DD showed atypical speech sound perception. I suggest that the author goes through the manuscript thoroughly and finds terms that clearly differentiate between categorization abilities and "mere" speech perception.

page 2, line 76: Why should atypical categorical perception be a precursor of dyslexia, if speech perception was deficient in FR children before formal instruction. I can see what the author is trying to say, but this should be made more explicit.

page 3 and 4: It seems somehow incomplete to only cite the Dutch and Finnish studies. The literature review should be broader.

page 5, line 228: what is the N-CDI - this needs to be explained when first mentioning it. Further, it sounds a bit "weird" that one infant was excluded for not filling in the N-CDI. Surely, the author means that parents did not fill in the parental questionnaire. Mentioning and explaining the N-CDI later is not helpful.

page 5, line 231: The author really needs to explain N-CDI before describing how it was used for exclusion and matching of participants. Further, I was wondering why the author only used the productive score. Especially, when considering that the experiment was on perception and not production. This needs to be explained and justified.

page 5, line 235: How come that there were more TD infants compared to FR infants, if they were matched? This is confusing, especially when looking at Table 1, where only 35 TD infants are included.

page 6, line 271: For the single-speaker condition: were the chosen tokens always the same for each infant or did the author use different tokens for different infants? This did not become clear.

page 8, line 342: Was down-sampling done before or after the filtering?

page 8, line 343: Using average re-referencing is not advisable for electrode montages with less than 64 electrodes. Therefore, I strongly encourage to use linked mastoids as an offline reference. Specifically, because amplitudes in central regions are reduced with the average reference, which, however, is one of the main regions of the MMN.

page 8, line 350: Having only 25 deviant trials as inclusion criterion is very low, considering that there were more than 100 deviants presented. At least 50% should be included. The author needs to justify and explain.

page 8, line 355: Only using 3 electrodes for analysis is not sufficient. As the author states him-/her-self, the MMN is fronto-centrally distributed. Therefore, at least the central electrodes should also be included. Another, better possibility would be to perform cluster-based permutation tests to identify the electrodes and time-window in which a significant MMR occurs. This would then also apply to how the author defined/can define the time windows.

Statistical analyses: As far as I understood, the author used the MMR difference wave to compare TD and FR infants. However, from my point of view, the author should perform a mixed-model ANOVA, in which it can be tested whether there is a significant stimulus (deviant, standard) x group interaction. If this was the case, the author can then compare the groups on the MMR difference wave. This is actually described later in the results section.

page 8, line 370: Did groups differ concerning accepted trials? This needs to be stated.

Figure 2 should be optimized. It would be a lot more helpful if there was a figure directly showing potential differences between groups.

page 11, line 415: The description of the results of the mixed-effect ANOVA are very difficult to follow. The only interesting effects are those including deviant and standard. All the other main effects and interactions can be left out or included in the supplements. The author is looking for an interaction involving ERPs (deviant, standard), condition (single, multiple speaker) and group, right? However, from what I read, this interaction did not become significant. Which, from my point of view, means that there were no group differences at all? This needs clarification.

page 12, line 452: I do not think that one can speak of different distribution, if two adjacent vs. only one electrode became significant. This should be excluded.

Results: Taken together, the author should reconsider analysis strategies (also preprocessing of data) and statistical analysis. I am not sure whether the statistical analysis supports conclusions the author draws.

Discussion:
1) the author suggests that the FR children might be at the transitional phase from p- to n-MMR in the single-speaker condition. Did the author check for this, at least descriptively? Otherwise, this is only speculation. This speculation can however be reduced if the author looked at violin diagrams or similar.

2) Discussing the "distribution" is not acceptable, when considering that effects were at one vs. two electrodes.

3) The author should put more emphasis on what the results would mean in terms of theories on the explanation of developmental dyslexia. Theories were not mentioned in this manuscript.

Reviewer 4 Report

Later but not weaker: neural categorization of native vowels of 2 children at familial risk of dyslexia

Ao Chen

The aim of the paper was to test 20 months-aged children on their neural categorization of acoustically similar native vowel contrast /i/ and /I/ realized in two nonwords, in order to differentiate asymptomatic children at risk of developing dyslexia and control children. Passive odd ball paradigm was recorded by typically developing (TD) children and children at familial risk of dyslexia (FR). When  2 nonwords s “giep” [É£ip] and “gip” [É£Ip] were pronounced in a multiple-speaker condition, significant infant positive mismatch response was elicited among both the TDs and the FRs, , but the FR hand a longer delay to mismatch peak. In the condition with only a single-speaker, FRs showed no significant MMR (while TD children showed a p-MMR at 360 ms). The authors conclude that TDs and FRs may process differently phonetic differences, but that both can at the end categorize correctly phonemes.

It is a simple conducted experiment. The results are clear. The children were 20 months old, and interestingly, Chen paired each individual FR child with a TD child closely matched by vocabulary. This point is quite interesting. The objective evaluation of the parents with reading tests are useful for a precise group definition. I find globally the articles brings consistent data to the problematic.

I have some comments under.

General

I have a general comment about authorship. I am surprised that this is one person’s study. In my experience one person cannot do an EEG study with children alone. There are 9 persons in the Acknowledgement. Is really a one person’s study? It is just a question, so if the author has good arguments, it is fine with me  

Introduction

 Concerning the important point: children with a dyslexic parent 29 have a 29% to 66% risk of developing dyslexia:

In my knowledge, there are two types of developmental dyslexia:  Phonological dyslexia, were children have problems with phonological awareness, + some visual problems, and surface dyslexia were children have difficulties in reading irregular words. It would be good to tackle this point (unless it is not pertinent), and discuss if the author tackles any type of dyslexia, or only one, and why.  

Method:

I have no comments on the stimuli and the presentation, which is done on a classical and rigorous way: The EEG was recorded in a sound-attenuated room at XXXX University. Which university?  I have a remark on statistical analysis, which is mentioned in the next paragraph (results)

An important point for me is the question of a control condition. It is not clear to me if it would have been important to have a control of a nonlinguistic mismatch (Tone? Blank noise), in order to have a more convincing explanation of the underlying mechanism. This is maybe not necessary, but this should be argued.

Ethic committee is correctly mentioned

Results

Thanks for these interesting results

Statistical analysis for EEG:  I am slightly confused by the EEG analysis, maybe it is a lack of habit with this specific technique. However, it is not clear to me if many electrodes  (like in the figure 2) were analyzed, or only F3, Fz, and F4 as suggested at a certain moment in the text. How many electrodes were analyzed in the scalp? Please complete the description also to improve description of first order corrections (Bonferroni or other) if needed.

Another point, which would improve the paper, would be to know what percentage of the FR children do show different EEG. Is this difference driven by a minority of infants, or is it a general trend of all infants?  

Discussion.

The discussion is interesting and adequate

The main point to tackle still is maybe the delay of p-MMR  In addition to the “classical” latency of M-MMR  is from what I know, either earlier (approx. 150–250 ms after the onset of the deviant stimulus), or later  latency range of 400–500 ms, but has also been observed at much later latencies. Here the situation is in between. Is that a question of age? Could the author discuss this point ?